# Structure of orthoreovirus RNA chaperone σNS, a component of viral replication factories

Boyang Zhao[1], Liya Hu[2], Soni Kaundal[2], Neetu Neetu[2], Christopher H. Lee[3,4], Xayathed Somoulay[3,4], Banumathi Sankaran [5], Gwen M. Taylor [4,6], Terence S. Dermody [3,4,6] ✉ & B. V. Venkataram Prasad [1,2] ✉

The mammalian orthoreovirus (reovirus) σNS protein is required for formation of replication compartments that support viral genome replication and capsid assembly. Despite its functional importance, a mechanistic understanding of σNS is lacking. We conducted structural and biochemical analyses of a σNS mutant that forms dimers instead of the higher-order oligomers formed by wildtype (WT) σNS. The crystal structure shows that dimers interact with each other using N-terminal arms to form a helical assembly resembling WT σNS filaments in complex with RNA observed using cryo-EM. The interior of the helical assembly is of appropriate diameter to bind RNA. The helical assembly is disrupted by bile acids, which bind to the same site as the N-terminal arm. This finding suggests that the N-terminal arm functions in conferring context-dependent oligomeric states of σNS, which is supported by the structure of σNS lacking an N-terminal arm. We further observed that σNS has RNA chaperone activity likely essential for presenting mRNA to the viral polymerase for genome replication. This activity is reduced by bile acids and abolished by N-terminal arm deletion, suggesting that the activity requires formation of σNS oligomers. Our studies provide structural and mechanistic insights into the function of σNS in reovirus replication.

Most viruses that replicate in the cytoplasm of host cells form neoorganelles that serve as sites of viral genome replication and particle assembly[1-3]. These highly specialized viral factory (VF) structures concentrate viral replication proteins and nucleic acids, limit activation of cell-intrinsic defenses, and coordinate release of progeny particles. Both viral and cellular proteins contribute to VF formation and orchestrate viral replication functions.

Mammalian orthoreoviruses (reoviruses) are members of the Orthoreovirus genus in the *Spinoreoviridae* family[4]. Reoviruses are

classified into four serotypes based on sequence and antigenicity of the σ1 viral attachment protein and include prototype strains type 1 Lang, type 2 Jones, type 3 Dearing, and type 4 Ndelle[5]. While reovirus can infect many mammalian species, including humans, symptomatic infection develops only in neonatal mammals and human infants and children[6]. Infection by some reovirus strains appears to induce the development of celiac disease, a complex immune disorder characterized by an immune response to dietary gluten that can damage the intestinal lining and result in diarrhea and malabsorption[7]. Reovirus can

[1]Department of Molecular Virology and Microbiology, Baylor College of Medicine, Houston, TX, USA. [2]Verna and Marrs Mclean Department of Biochemistry and Molecular Pharmacology, Baylor College of Medicine, Houston, TX, USA. [3]Department of Microbiology and Molecular Genetics, University of Pittsburgh School of Medicine, Pittsburgh, PA, USA. [4]Institute of Infection, Inflammation, and Immunity, UPMC Children's Hospital of Pittsburgh, Pittsburgh, PA, USA. [5]Berkeley Center for Structural Biology, Molecular Biophysics and Integrated Bioimaging, Lawrence Berkeley Laboratory, Berkeley, CA, USA. [6]Department of Pediatrics, University of Pittsburgh School of Medicine, Pittsburg, PA, USA. ✉e-mail: terence.dermody@chp.edu; vprasad@bcm.edu

induce apoptosis, necroptosis, and pyroptosis in tumor cells by activating programmed cell death pathways[8]. Because of these characteristics, reovirus was one of the first viruses used as an oncolytic agent[9].

Reovirus forms nonenveloped particles with an icosahedral capsid, 800 Å in diameter, consisting of two concentric protein layers and distinctive turret-like spikes projecting from the twelve icosahedral vertices. The viral genome is encapsidated within the innermost protein layer and consists of ten segments of linear double-stranded (ds) RNA, which encode eleven viral proteins. Eight of these proteins are structural and form the two capsid layers, turret, attachment protein, and an internally located RNA-dependent RNA polymerase. There are three nonstructural proteins that function to promote viral replication, which occurs in the cytoplasm. After binding to cellular receptors, the virus particle enters many types of cells by clathrin-mediated endocytosis[10]. Within endosomes, proteolysis of outer-capsid proteins yields infectious subvirion particles, which penetrate endosomal membranes to release transcriptionally active viral cores into the cytoplasm[11]. Following transcription of the dsRNA segments within the core interior, capped viral mRNAs are extruded through the turrets into the cytoplasm. The capped transcripts serve as templates for translation as well as synthesis of minus-strand genomic RNA by the viral polymerase to produce the dsRNA genome segments. Assortment of the genome segments into progeny core particles and the addition of outer-capsid proteins occur inside VFs. Nonstructural proteins σNS and μNS are required to nucleate these compartments during infection.

In addition to its essential function in the formation of viral replication factories, σNS recruits viral RNA[12] and both the eukaryotic translation initiation factor 3 subunit A (eIF3A) and the ribosomal subunit pS6R to enhance viral RNA translation[13]. Its association with the major stress granule effector protein, Ras-GAP SH3-binding protein 1 (G3BP1), disrupts stress granule formation, which relieves a block to reovirus replication[14]. σNS binds viral single-stranded (ss) RNA nonspecifically[15], protects ssRNAs from degradation, and ferries viral mRNAs to VFs[16]. A mechanistic understanding of this multifunctional protein is lacking without its atomic-resolution structure. Our previous studies using cryo-EM show that σNS forms heterogeneous, higher-order oligomers and, when associated with ssRNA, it forms filamentous structures[16]. However, neither the unliganded nor the RNA-bound form of σNS is conducive to high-resolution structural studies.

Here, we conducted structural and biochemical studies of σNS and discovered that domain-swapping interactions of the flexible N-terminal arms of σNS dimers facilitate the association of dimers into filamentous structures that can bind ssRNA. Moreover, we found that such assemblies are required for an RNA chaperone activity of σNS, which we postulate is required for reovirus replication. Bile acids bind to the same hydrophobic pocket as the N-terminal arms, disrupt filament formation, and impede RNA chaperone activity. Collectively, these findings provide a mechanistic understanding of σNS function in reovirus replication and provide clues about dsRNA synthesis in a *Spinoreoviridae* virus.

## Results

### An R6A mutation in σNS disrupts RNA-binding capacity and oligomerization

Initial attempts to crystalize wildtype (WT) σNS were unsuccessful due to the binding of the protein to RNA, which leads to the formation of higher-order oligomers that disassemble into heterogeneous populations during purification and crystallization. The N-terminal 38 residues of σNS include several amino acids required for the binding of σNS to RNA[16]. These residues are conserved in all reovirus σNS sequences reported to date[17]. We previously engineered σNS mutants incapable of binding RNA to assess the requirement of σNS RNA-binding activity in reovirus replication[12]. The σNS-R6A mutant does not distribute to VFs and fails to transport viral mRNA to these structures[12]. Based on its lack of RNA-binding capacity, we selected the σNS-R6A mutant for purification and crystallization. Expression and purification of σNS-R6A yielded pure and homogenous recombinant protein. Following anion-exchange chromatography, we compared recombinant WT σNS and σNS-R6A for interactions with nucleic acid (Fig. 1a). Anion-exchange chromatography of WT σNS showed two peaks in contrast to that of the σNS-R6A mutant, which showed only one peak, consistent with the failure of σNS-R6A to bind RNA. Size-exclusion chromatography of WT σNS following anion-exchange chromatography, yielded an oligomer of a size corresponding to a decamer (Fig. 1b) in contrast to σNS-R6A, which showed a peak corresponding to a dimer (~80 kDa). Surprisingly, the selenomethionine (Se-Met)-substituted σNS-R6A, which we purified for crystallization attempts, showed a size-exclusion chromatography profile suggesting a multimeric association with a peak corresponding to an octamer.

### Crystal structure of σNS-R6A

We determined the structure of the σNS-R6A mutant by Se-Met single-wavelength anomalous diffraction (SAD) phasing to a resolution of 3.0 Å (Table 1). The crystals were in the space group P6₅ with two molecules in the asymmetric unit. The σNS-R6A monomer structure consists of a protruding arm formed by the N-terminal 17 residues that connect to the globular core consisting of several β-strands and α-helices (Fig. 2a, b). The antiparallel dimer in the asymmetric unit is stabilized by extensive interactions between the subunit cores with the N-terminal arms protruding on either side (Fig. 3a, b). The dimeric interface consists of a combination of hydrophobic residues (Gly 148 and Val 160) and polar residues (Arg 67, Arg 159, His 164, Asp 168, and

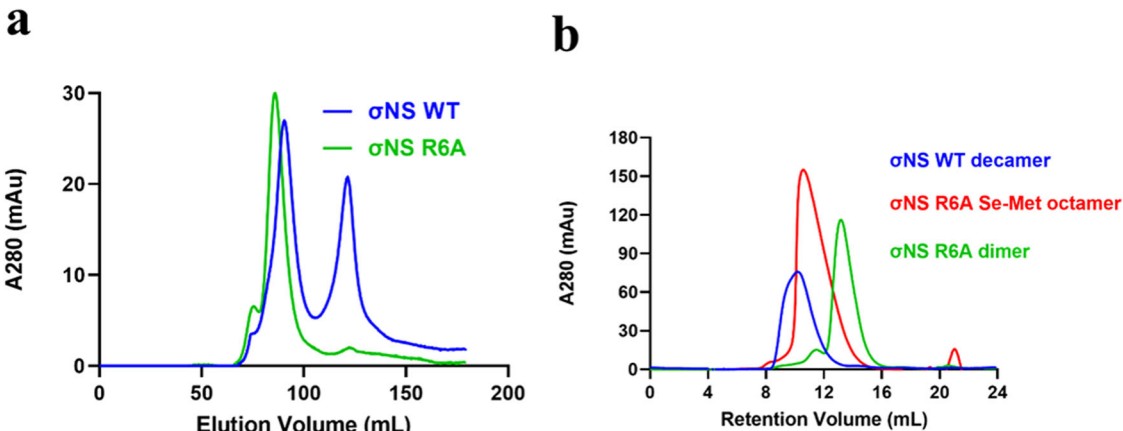

**Fig. 1 | Purification of σNS-R6A dimer.** Elution profiles of WT σNS and σNS-R6A after **a** anion-exchange chromatography and **b** size-exclusion chromatography using an S200 16/60-increase column. A size shift is observed with the different elution volumes for each peak. See Fig. S1 for purified WT σNS and σNS-R6A absorption spectra.

**Table 1 | X-ray diffraction data and structure refinement statistics**

| PDB ID | 8TKA | 8TL8 | 8TL1 |
|---|---|---|---|
| Wavelength (Å) | 0.99 | 0.99 | 0.99 |
| Resolution range (Å) | 34.28–3.0 (3.10–3.0) | 34.22–3.20 (3.31–3.20) | 34.52–3.16 (3.27–3.16) |
| Space group | P 6₅ | P 4₁ | C 1 2 1 |
| Unit cell | | | |
| a, b, c (Å) | 137.11, 137.11, 98.24 | 54.12, 54.12, 305.52 | 77.52, 76.80, 89.09 |
| α, β, γ (°) | 90, 90, 120 | 90, 90, 90 | 90, 94.27, 90 |
| Unique reflections | 41331 (4152) | 11801 (1150) | 7841 (302) |
| Multiplicity | 2.0 (2.0) | 1.0 (1.0) | 1.0 (1.0) |
| Completeness (%) | 99.77 (99.67) | 81.91 (82.38) | 86.39 (33.82) |
| Mean I/σ(I) | 21.07 (3.53) | 18.04 (5.40) | 10.09 (3.21) |
| Wilson B-factor (Å²) | 68.54 | 62.51 | 67.74 |
| R-merge | 0.07 | 0.03 | 0.13 |
| R-work | 0.24 (0.33) | 0.22 (0.29) | 0.208 (0.25) |
| R-free | 0.29 (0.41) | 0.24 (0.29) | 0.250 (0.28) |
| Number of non-hydrogen atoms | 5494 | 5337 | 2708 |
| Macromolecules | 5494 | 5205 | 2637 |
| Ligands | 0 | 132 | 66 |
| Solvent | 0 | 0 | 5 |
| Protein residues | 708 | 669 | 336 |
| R.M.S. deviations | | | |
| Bond lengths (Å) | 0.012 | 0.005 | 0.010 |
| Bond angles (°) | 1.60 | 1.06 | 1.36 |
| Ramachandran | | | |
| Favored (%) | 96.29 | 97.43 | 95.48 |
| Allowed (%) | 3.14 | 2.57 | 3.61 |
| Disallowed (%) | 0.57 | 0.00 | 0.90 |
| Average B-factor (Å²) | 81.30 | 59.23 | 67.73 |
| Macromolecules | 81.30 | 58.98 | 67.10 |
| Ligand | | 69.46 | 95.72 |
| Solvent | | | 30.00 |

Numbers in parentheses correspond to the highest resolution range.

Glu 243) that engage in hydrogen bond interactions (Fig. 3c). The calculated buried surface area (BSA) of 1853 Å at the dimeric interface indicates a stable dimer, which is consistent with the observation that σNS-R6A exists as a dimer in solution.

## Formation of a helical assembly involving domain-swapping N-terminal arm interactions

The σNS-R6A dimers in the asymmetric unit associate to form a helical assembly extending from one unit cell to the next along the *c* axis of the crystal, with the non-crystallographic axis, relating the two molecules, approximately perpendicular to the helix axis. The N-terminal arms of the two subunits project laterally, away from the body of the subunit, and insert into grooves in the neighboring dimers, stabilizing the helical assembly with an overall diameter of ~150 Å and a 40 Å central cavity (Fig. 4a–d). The formation of such a helical assembly is consistent with previous observations using cryo-EM that σNS forms filamentous structures of similar diameter in the presence of RNA[16]. Helical assembly is not expected for the σNS-R6A mutant, as it does not bind RNA and forms only dimers in solution. However, the size-exclusion chromatography profile of Se-Met-substituted σNS-R6A clearly showed that it forms higher-order oligomers in solution, as observed with WT σNS (Fig. 1b). It is possible that increased hydrophobicity due to substitution of a selenium atom for a sulfur atom[18] in Met 1 coupled with crystal-packing forces restricts the flexibility of the N-terminal arm and allows this region of the protein to stabilize a multimeric association. We used molecular dynamics simulations based on the σNS-R6A structure to examine the flexibility of the N-terminal arm in σNS and determine how this flexibility is affected by the presence of RNA and the Se-Met substitution in σNS-R6A. These studies showed that the N-terminal arm in σNS is significantly more flexible relative to other parts of the protein and that this flexibility is hindered by the addition of RNA (Fig. S2a). In addition, Se-Met substitution in the σNS-R6A mutant reduces the flexibility of the N-terminal arm and stabilizes dimer-dimer interactions (Fig. S2b, c).

## Bile acid derivatives disrupt σNS filament formation

We observed that the native σNS-R6A crystals, without Se-Met substitution, crystallized in the P4₁ space group instead of P6₅. Native σNS-R6A crystallized only in conditions with bile acid derivatives in contrast to Se-Met σNS-R6A, which crystallized without these additives. Structure determination of native σNS-R6A, using the Se-Met σNS-R6A structure as a molecular replacement model, showed that native σNS-R6A also forms an antiparallel dimer, with two bile acid moieties bound to each subunit in a groove located near the σNS C-terminus (Fig. 5a, sites 1 and 2). Binding of bile acid molecules is mediated by hydrophobic contacts with residues Thr 107, Glu 110, Leu 111, Ser 114, Gly 203, Leu 204, Tyr 240, Glu 242, Ala 245, and Glu 246 (Fig. 5b). In contrast to the Se-Met σNS-R6A structure, the N-terminal arm is disordered and not observed in the native σNS-R6A structure. Instead of the distinct helical assembly in the structure of Se-Met σNS-R6A, native σNS-R6A dimers crystallized with bile acid show sheet-like packing (Fig. 5c). Further examination of the structure showed that one of the bile acid molecules occupies the same location in the core domain as the N-terminal arm required to form helical assemblies, suggesting that bile acid competes with the N-terminal arm to disrupt the helical assembly formation observed in the Se-Met σNS-R6A structure (Fig. 5d–f).

## The σNS N-terminal arm is required for the formation of σNS multimers

To provide additional evidence for the role of domain-swapping interactions of the N-terminal arms in forming oligomeric assemblies, we engineered a deletion construct, σNS-ΔN17, by truncating the N-terminal 17 residues of the protein. The σNS-ΔN17 mutant formed dimers in solution and also crystallized in the presence of bile acid derivatives. The σNS-ΔN17 crystal structure determined using molecular replacement showed a dimer formed by a central core identical to that in the structures of Se-Met or native σNS-R6A (Fig. 5g) with two bound bile acid moieties. As expected, no helical packing was observed in the crystal structure. Therefore, the structure of σNS-ΔN17 suggests that the N-terminal arm and the core of WT σNS operate as independent structural units, and removing the N-terminal arm does not alter the dimeric association of the core. These three crystal structures provide strong evidence that (i) WT σNS forms dimers, (ii) the dimer is the basic unit of multimeric assembly, and (iii) domain-swapping interactions of the flexible N-terminal arms are required to link the dimeric units to form a multimer. Furthermore, the σNS-ΔN17 structure showed that bile acid derivatives bind to the same location in σNS as the N-terminal arm in the σNS-R6A structure (Fig. 5g), suggesting that this location in the σNS core domain has an affinity for cholesterol-like molecules and that binding to such molecules disrupts filamentous helical assemblies by competing with the binding of the N-terminal arms.

## σNS oligomer has RNA chaperone activity

The surface representation of the helical assembly in the Se-Met σNS-R6A structure shows a negatively charged exterior (Fig. 6a) and a

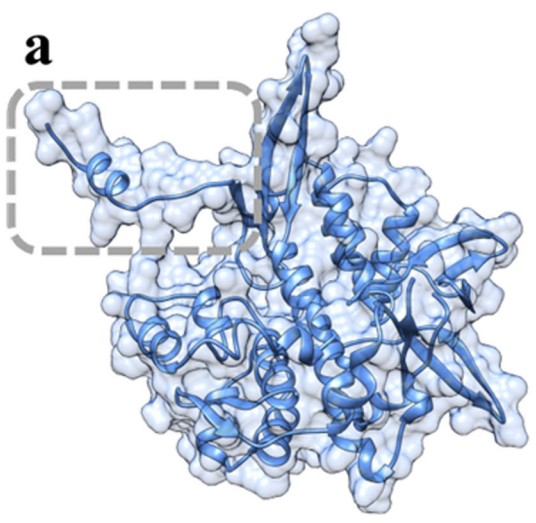

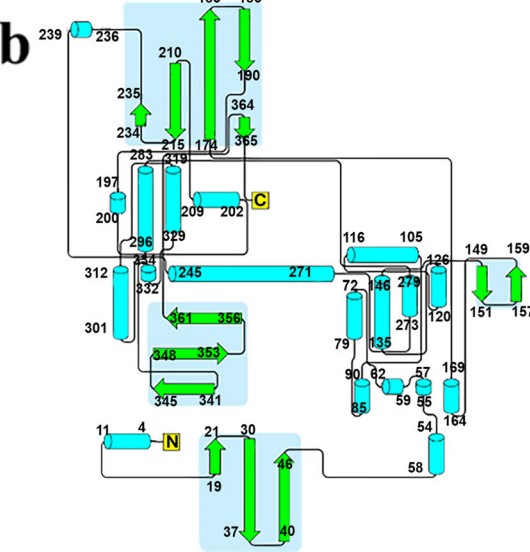

**Fig. 2 | The crystal structure of the σNS R6A mutant contains an N-terminal arm extending from a globular core. a** Ribbon and surface representations of the σNS crystal structure. The N-terminal arm protruding from the globular core is demarcated in a grey-dotted frame. **b** Topologic representation of the σNS R6A structure generated by the PDBsum database, β-strands colored in green with a light blue background, α-helices colored in cyan, and the connecting loops colored in grey.

positively charged interior (Fig. 6b). A cutaway view of the central tunnel of the helical assembly reveals a substantial number of positively charged residues, suggesting that this region can bind RNA by electrostatic interactions. In silico docking experiments demonstrated that the central tunnel could accommodate an RNA strand surrounded by σNS dimers (Fig. 6c–e). This finding, coupled with previous studies showing that avian reovirus σNS has RNA chaperone activity[19], prompted us to hypothesize that mammalian orthoreovirus σNS is likewise an RNA chaperone. To test this hypothesis, we engineered an RNA helix substrate containing 5′ and 3′ single-stranded 6-nuclotide overhangs by annealing a short hexachloro-fluorescein (HEX)-labeled RNA with a longer unlabeled RNA (Fig. 6f and Table S1). To first examine whether purified WT σNS and σNS-R6A have RNA helix-destabilizing activity, we incubated each of these proteins with the RNA helix substrate. Following incubation, the reaction mixtures were treated with proteinase K and resolved by electrophoresis in nondenaturing polyacrylamide gels. The RNA-RNA duplex migrated as a single prominent band when σNS was not added to the reaction mixtures (Fig. 6h and S2a). However, following the addition of increasing concentrations of WT σNS, we observed a proportional increase in the formation of ssRNA molecules. In contrast, the formation of ssRNA was not observed following the addition of σNS-R6A (Fig. 6i and S2b). These results indicate that σNS has RNA-destabilizing activity, which is abrogated by the R6A mutation. We next evaluated whether σNS is capable of annealing two ssRNA molecules to form dsRNA. For this experiment, we engineered two 42-nucleotide complementary RNA strands with a defined stem-loop secondary structure. One of these strands was 5′ HEX labeled, while the other was not, as shown in Fig. 6g. We observed a proportional increase in the formation of dsRNA when equal amounts of the two ssRNA strands were mixed and incubated with increasing concentrations of WT σNS (Fig. 6j and S2c), indicating that σNS facilitates the formation of dsRNA molecules from ssRNA and thus has RNA-annealing activity. Since the reaction buffer did not contain ATP, the RNA helix-destabilizing activity of σNS is independent of energy input, indicating that σNS is not a helicase but an RNA chaperone.

## Binding of bile acid inhibits σNS chaperone activity

From our observation that bile acid binding disrupts the formation of filamentous helical assemblies of σNS, we hypothesized that bile acid binding would impede σNS RNA chaperone activity. To test this hypothesis, we conducted RNA helix-destabilization and -annealing assays following incubation of σNS with different components of bile acid derivatives, including taurocholic acid sodium salt hydrate (TASAH), sodium glycocholate hydrate (SGAH), 3-[(3-cholamido-propyl)dimethylammonio]−1-propanesulfonate (CHAPS), and 3-[(3-cholamidopropyl)dimethylammonio]−2-hydroxy-1-propanesulfonate (CHAPSO). Each of these components reduced the RNA-destabilizing activity (Fig. 6k and S2d) and RNA strand-annealing activity (Fig. 6l and S2e) of σNS to varying degrees, suggesting that the RNA chaperone activity of σNS requires the formation of filament-like assemblies. As anticipated, RNA-binding capacity and oligomer formation of σNS were abolished by the removal of the N-terminal 17 residues and, concomitantly, the σNS-ΔN17 mutant did not have RNA chaperone activity (Fig. 6m and S2f).

## Discussion

VFs serve an essential function in the replication of many viruses, as these structures house viral genome synthesis and assembly of progeny virions. In the case of mammalian orthoreoviruses, two nonstructural proteins, σNS and μNS, initiate VF formation, and σNS recruits viral RNAs into these compartments[12]. Our crystallographic and biochemical studies reported here provide a mechanistic understanding of how σNS forms filamentous assemblies by linking dimeric units through domain-swapping interactions of the N-terminal arms and further how such a multimeric association is required for the RNA helix-destabilizing and RNA-annealing activities of σNS. Previous cryo-EM studies show that in the presence of RNA σNS forms filamentous structures of similar dimensions as observed in our crystallographic studies[16]. These findings suggest that σNS unwinds the viral RNA for delivery to the viral RNA-dependent RNA polymerase during genome replication and promotes interactions between the viral RNAs to facilitate their selective encapsidation during virion assembly.

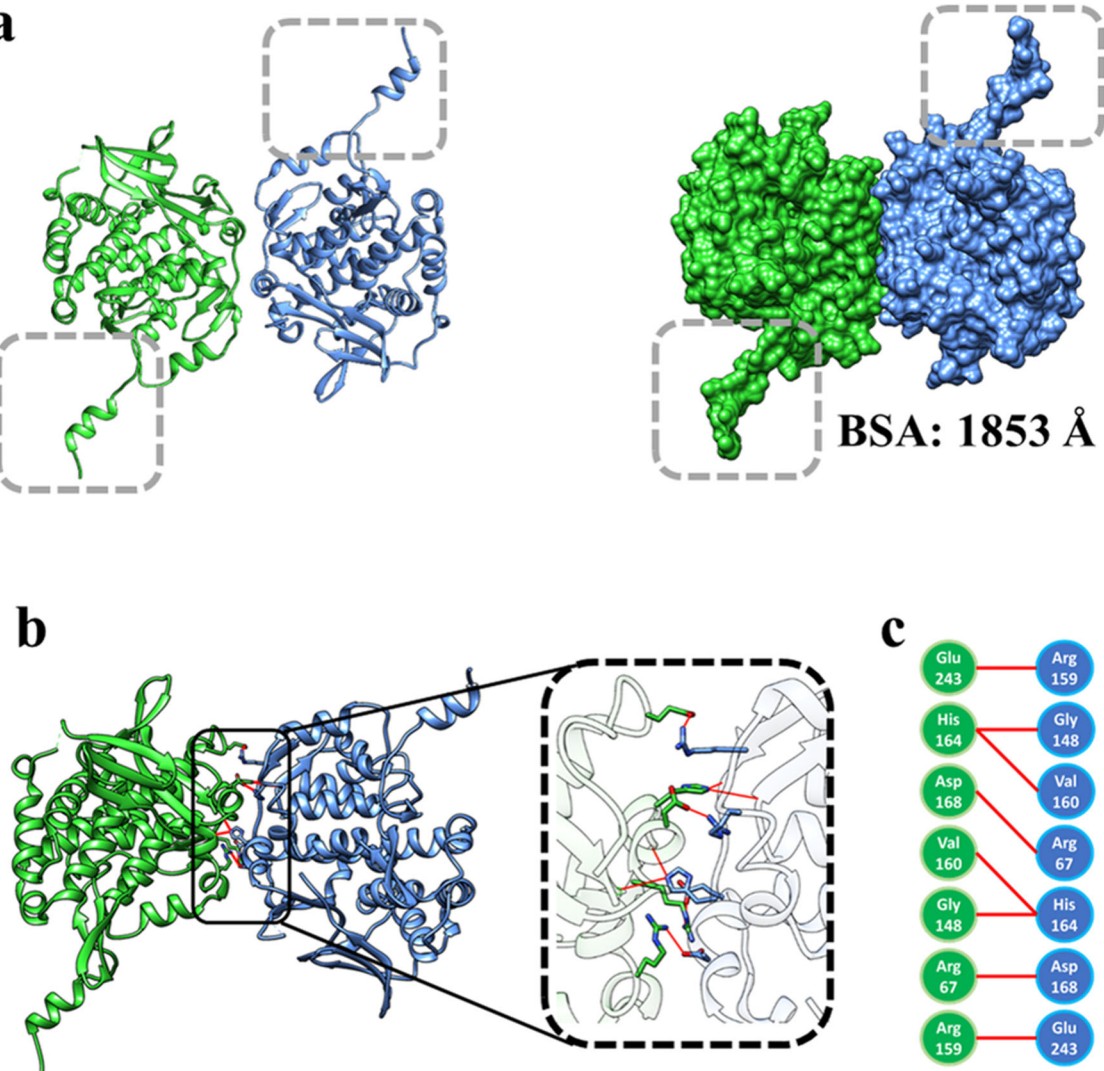

**Fig. 3 | Antiparallel σNS R6A dimer in the crystallographic asymmetric unit.**
**a** Ribbon (left) and surface representations (right) of the σNS R6A dimer with subunits colored in green and blue. The N-terminal arms are demarcated in grey-dotted frames. **b** Ribbon representation of the σNS R6A dimer highlighting the residues at the dimeric interface inside a black frame in blue and green. **c** Amino acid residues from the subunits at the dimer interface are colored blue and green. Hydrogen bond interactions are shown by red lines.

## Higher-order oligomers and a flexible element—a common theme in VF formation?

Although distinct in its structure, σNS has several functional and mechanistic similarities with other dsRNA virus nonstructural proteins, such as rotavirus NSP2 and rice black-streaked dwarf virus (RBSDV) P9–1. Like σNS, both NSP2 and P9-1 are implicated in the formation of VFs. The striking common structural feature of these proteins is the formation of oligomeric assemblies requiring a flexible structural element. However, they differ in detail. Unlike the formation of filamentous assemblies by σNS, rotavirus NSP2 forms a donut-shaped octamer[20,21]. Instead of the flexible N-terminal arm of σNS, which is responsible for σNS dimer-dimer association in filamentous assembly formation, NSP2 has a flexible C-terminal arm, which facilitates inter-octamer association under crystallographic conditions through domain-swapping interactions with neighboring NSP2 molecules[20]. Inter-octamer association is thought to be the basis for VF formation during rotavirus replication[20,22]. RBSDV P9-1 also forms an octamer using a flexible C-terminal arm[23–25]. Like rotavirus NSP2, higher-order P9-1 octamers are required for the formation of VFs, as deletion of this sequence disrupts RBSDV VFs[25]. From our studies, deletion of the N-terminal arm of σNS abrogates dimer-dimer interactions and precludes the formation of higher-order multimers. However, further experiments are required to assess the function of the flexible N-terminal arm in the formation of functional VFs.

## RNA chaperone activity—a flexible N-terminal arm

An important finding from our studies is that mammalian orthoreovirus σNS has RNA helix-destabilizing and RNA-annealing activities without requiring a metal ion or ATP, which makes this protein an RNA chaperone[26]. Similar RNA chaperone activity has been reported for rotavirus NSP2[27] and avian reovirus σNS[19]. The RNA duplex used in our experiments contains overhangs at the ends of both strands, suggesting that passive strand displacement activity is initiated after σNS binds to unduplexed overhang regions. The binding of σNS would then prevent reannealing of the duplex and further separate the strands. Using complementary ssRNAs, our studies show that σNS binds, unfolds, and releases these molecules to facilitate the formation of duplex RNA.

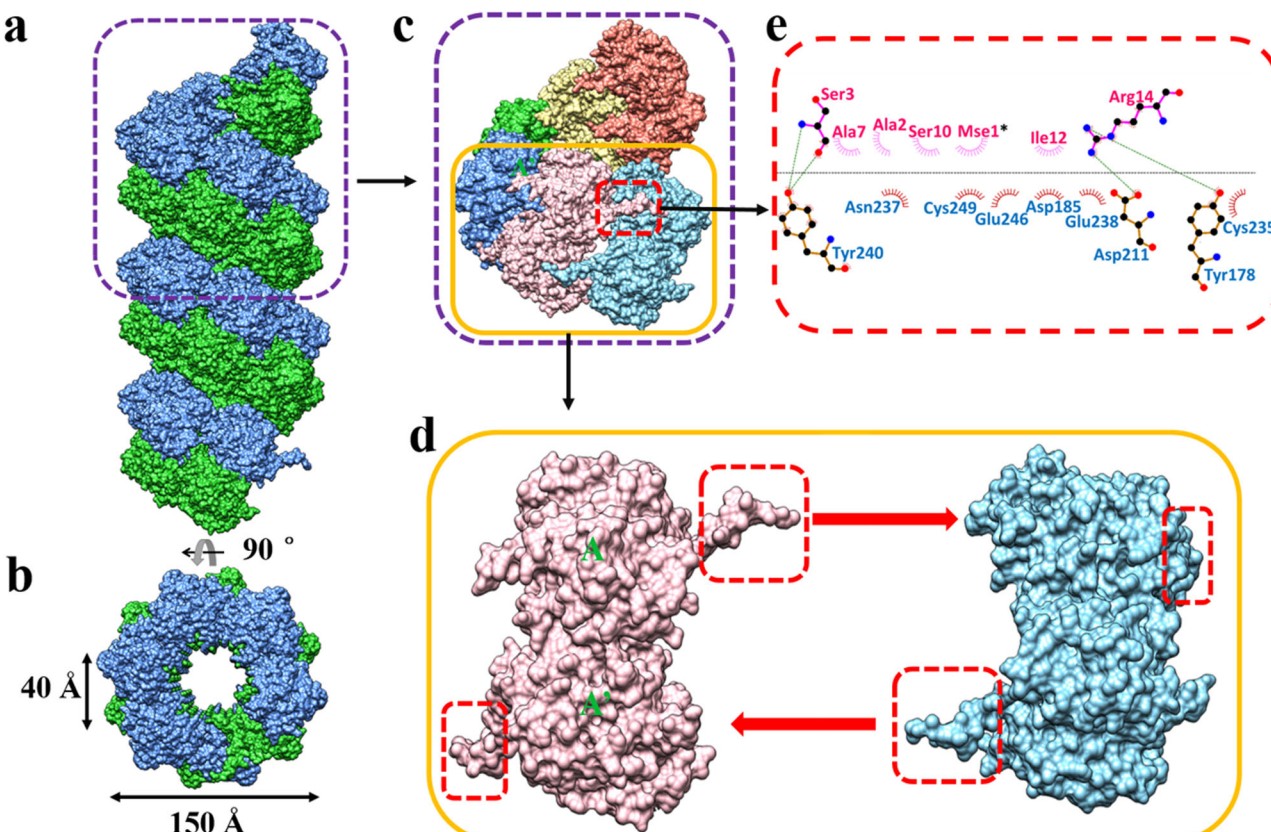

**Fig. 4 | Formation of σNS filaments. a** Surface representation of the helical assembly formed by σNS dimers using crystallographic P6$_5$ symmetry along the crystallographic *c* axis. Dimers are depicted as blue and green subunits. A single helical turn (80 Å in length) is demarcated in the purple-dotted frame. **b** A 90° rotation around the horizontal axis perpendicular to the filament shows the central tunnel. The tunnel is 40 Å in width, and the total diameter of the filament is 150 Å. **c** Single helical turn formed by six σNS dimers. Dimers are depicted in different colors. One pair of interacting dimers (pink and cyan) are demarcated in a yellow frame. The location of the N-terminal arm of the pink dimer interacting with the cyan dimer is shown within the red frame. **d** Interacting dimers (pink and blue, two monomeric subunits in the pink dimer denoted A and A') corresponding to the yellow frame in **c** are shown separately to illustrate how the N-terminal arms (in red frame) project away in opposite directions (thick red arrows) to chain-link the neighboring dimers to form the helical assembly shown in Fig. 4a. **e** Ligplot of interactions between N-terminal arm residues of the pink dimer and core domain residues of the blue dimer. Residues involved in hydrophobic interactions are shown by brick-red and pink spoked arcs, and hydrogen bond interactions between the residues are indicated by green lines, with carbon, oxygen, and nitrogen atoms depicted in black, red, and blue, respectively. Note the involvement of the Se-Met residue at position 1 (Mse 1), indicated by an asterisk.

The RNA chaperone activity of σNS requires the flexible N-terminal arm with arginine at residue 6 to function in domain-swapping interactions, as the R6A substitution or deletion of the N-terminal arm curtails this activity. In an analogous scenario, deletion of the flexible C-terminal arm of NSP2 also significantly reduces RNA chaperone activity[28]. Thus, the requirement for a flexible region in these proteins suggests that the mechanism is likely mediated by disorder or entropy transfer, as suggested for other viral RNA chaperones[29]. One possibility is that σNS uses the flexibility of the N-terminal arm for binding RNA by promoting oligomerization and subsequently releasing the bound RNA. Such a possibility is consistent with our observation that a single residue, Arg 6, in the N-terminal arm influences the RNA-binding capacity of σNS[12] by directly interacting with the RNA, disrupting σNS oligomerization, or both. The free N-terminal arms of the σNS multimer subunits (Fig. 5e) may exist in a dynamic equilibrium between a freely exposed open state, in which Arg 6 can initiate binding to RNA, and a closed state, in which the N-terminal arm is engaged in domain-swapping interactions for RNA release. Such dynamics also may allow the N-terminal arms to recruit free σNS dimers to elongate the multimer and coat the RNA as a mechanism for unwinding (Fig. 7). The dynamics and preference between open and closed states are likely influenced by the presence of RNA and buffer conditions (pH and ionic strength). This idea is

consistent with our observation that in solution, σNS exists as an octamer in the absence of RNA, but in the presence of RNA, as observed in our previous cryo-EM images[16], σNS forms filamentous structures of varying lengths.

### Bile acid binding–tampering with the flexibility of the N-terminal arm

A fascinating finding from our study is that bile acid derivatives bind to the same site in σNS as the N-terminal arm and impede the RNA chaperone activity of σNS (Fig. 5h). These findings are in agreement with the proposed mechanism of σNS RNA chaperone activity. First, the σNS crystal structure shows that the free N-terminal arm displaced by bile acid is disordered, as the electron density for this region is absent in the presence of bile acid. This observation is consistent with the disorder or entropy-transfer model of RNA chaperone activity[29], in which entropy is transferred to bound RNA to facilitate RNA refolding. Second, our finding that bile acid binding does not entirely abrogate RNA chaperone activity concurs with the idea that RNA modulates the kinetics of open and closed states of the N-terminal σNS arm (Fig. 7). It is possible that differences in the efficiency by which bile salts diminish RNA chaperone activity are determined by differences in the capacity of these molecules to compete with the N-terminal arm in the presence of RNA.

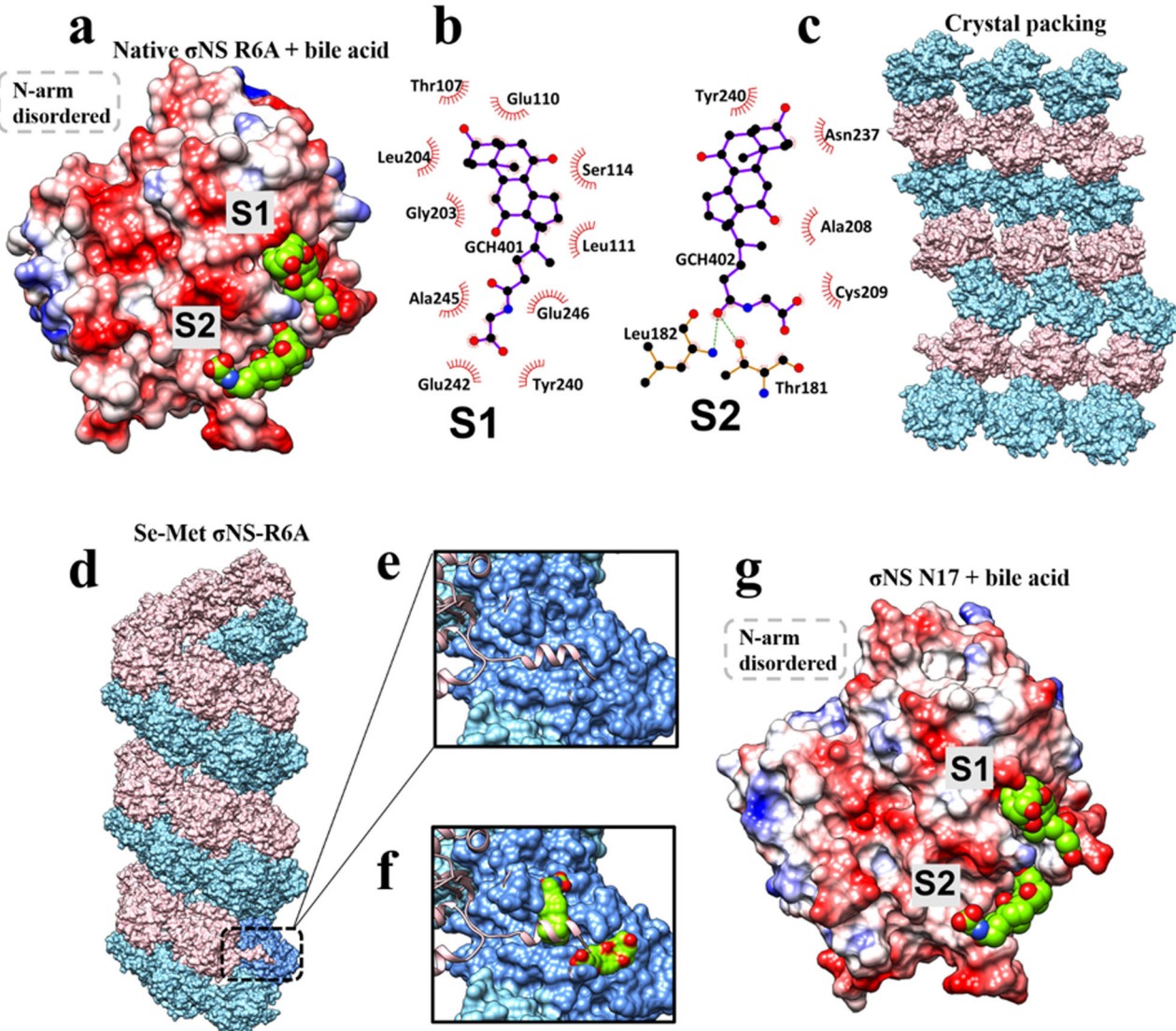

**Fig. 5 | Binding of bile acid derivatives disrupts σNS oligomerization and RNA-destabilizing activity. a** Electrostatic representation of σNS-R6A in complex with bile acid derivatives (S1 and S2), colored in green. The grey-dotted box demarcates the missing disordered N-terminal arm. **b, c** Residues mediating bile acid-derivative binding at S1 and S2 defined by LigPlot following the same notations as in Fig. 4e for hydrophobic and hydrogen bond interactions. Bile acid derivatives are shown in purple bonds with carbon and oxygen atoms in black and red, respectively. **c** The higher-order structure formed by crystal packing of σNS-R6A when bile acid derivatives are present. **d** The helical assembly as observed in the crystal structure of Se-Met σNS-R6A with dimeric subunits shown in pink and blue. **e** Close-up view of the N-terminal arm (pink) interacting with a groove in the neighboring subunit (blue). **f** In the structure of σNS-R6A in complex with bile acid derivatives, the same groove is occupied by a bile acid replacing the N-terminal arm, which becomes disordered. **g** Electrostatic representations σNS-ΔN17 in complex with two bile acid derivatives (in green) bound at the same location as in the σNS-R6A structure shown in Fig. 5a. The grey-dotted box demarcates the missing N-terminal arm.

## Role of RNA chaperone activity in reovirus replication

The RNA chaperone activity of σNS raises an important question about the stage in viral replication at which this activity is required. While the function of σNS RNA-binding activity in the recruitment of viral RNAs to VFs has been established[12], further studies are required to unequivocally determine the precise function of σNS RNA chaperone activity. As proposed for RNA chaperones of ssRNA viruses[29], σNS RNA chaperone activity may be necessary for unfolding viral mRNA for translation, replication, packaging, or some combination of these processes. In the case of dsRNA viruses like reoviruses and rotaviruses that package multiple dsRNA segments, RNA chaperone activity is likely essential for proper refolding of the kinetically trapped intermediate structures in capped viral mRNAs to allow these RNAs to serve as templates for dsRNA synthesis by the viral polymerase and also for facilitating interactions between specific viral RNAs for gene segment assortment and packaging, as proposed for reovirus σNS analog, rotavirus NSP2[30].

## Methods

### Expression and purification of σNS constructs

Synthesized genes for WT σNS (strain type 3 Dearing), σNS-R6A, and σNS-ΔN17 were subcloned into the bacterial expression vector pET28 with an N-terminal His tag and a TEV protease cleavage site (Epoch Life Science). *Escherichia coli* DE3 cells (Novagen) were transformed with the pET28 plasmid and induced with 0.5 mM isopropyl β-D-1-thiogalactopyranoside (IPTG) (Millipore Sigma) when the optical density at 600 nm reached 0.6. *E. coli* B834 cells were incubated in Dream medium supplemented to contain Dream Nutrient Mix and 10 mg/mL of Se-Met. Cells were resuspended in 50 mM Tris-HCl (pH 8.0), 300 mM NaCl, and 10 mM imidazole and supplemented to

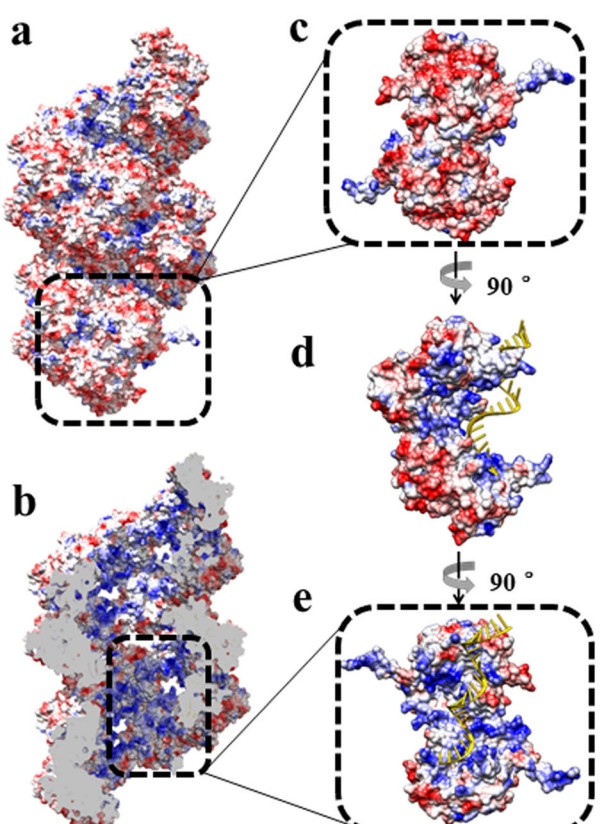
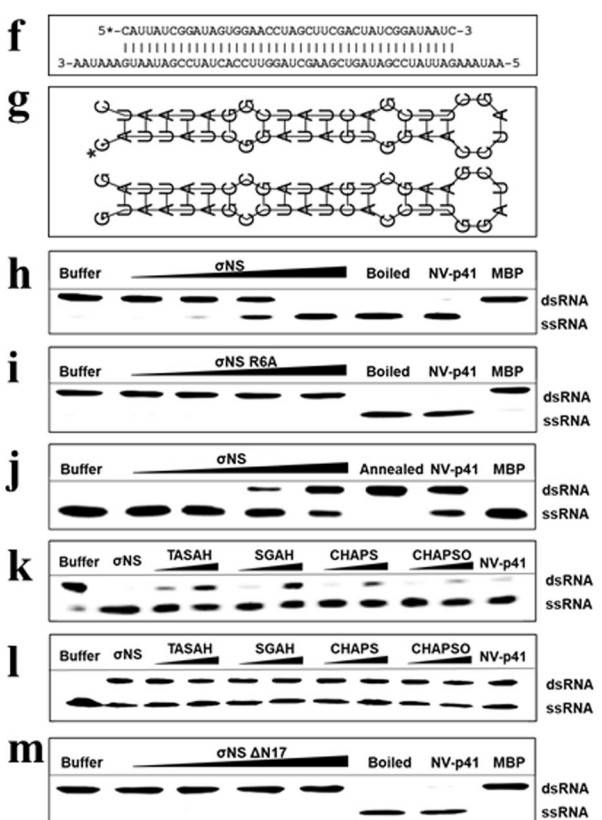

**Fig. 6 | RNA chaperone activity of σNS. a** Electrostatic representation of a σNS filament with red indicating negatively charged residues and blue indicating positively charged residues. The exterior of the filament is negatively charged. **b** Cutaway view of the filament. The interior of the filament is positively charged. **c** A single dimer is demarcated in a black-dotted frame with its relative position within the filament. **d** A 90-degree rotation around the *z* axis from the previous position shows a side view of the dimer. A docked RNA molecule is colored in yellow. **e** A 90-degree rotation around the *z* axis from the previous position shows the positively charged pocket of σNS that binds to a docked RNA molecule colored in yellow. The black-dotted frame depicts the position of the dimer within the filament. **f** Schematic representation of the RNA helix substrate containing overhangs at 3' and 5' ends. **g** Schematic representation of the stem-loop structures of the complementary 42-nucleotide RNA strands. The asterisk indicates the HEX-labeled

strand in **f** and **g**. **h** RNA helix-destabilizing activity of WT σNS. Increasing concentrations of WT σNS yield more ssRNA products. MBP and norovirus (NV) p41 were used as negative and positive controls, respectively. **i** Helix-destabilizing activity of σNS-R6A. **j** RNA-annealing activity of WT σNS. Increasing formation of dsRNA with increasing σNS concentration. Boiling followed by cooling to room temperature of the two ssRNA strands was used as a control. **k, l** Effects of bile acid derivatives on the RNA helix-destabilizing (**k**) and RNA strand-annealing activity (**l**) of σNS. TASAH, taurocholic acid sodium salt hydrate. SGAH, sodium glycocholate hydrate. CHAPS, 3-[(3-cholamidopropyl)dimethylammonio]-1-propanesulfonate. CHAPSO, 3-[(3-cholamidopropyl)dimethylammonio]−2-hydroxy-1-propanesulfonate. **m** RNA-helix destabilizing activity of σNS-ΔN17. Increasing concentrations of σNS-ΔN17 does not yield significant ssRNA products. All the experiments for 6h-m are done three times independently. Source data are provided as a Source Data file.

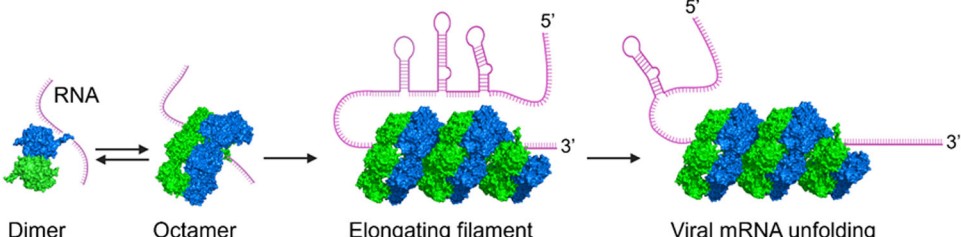

**Fig. 7 | Model of σNS oligomerization, RNA binding, and RNA unfolding.** The flexible N-terminal arms of σNS are required for RNA chaperone activity, consistent with an entropy-transfer model for RNA chaperones. The free N-terminal arms of σNS oligomers exist in equilibrium between an open state that can initiate RNA

binding and a closed state that engages in domain-swapping interactions and RNA release from the N-terminal arm. The dynamics of the N-terminal arms allow for the recruitment of free σNS dimers to elongate σNS oligomers to coat the RNA and enhance RNA unfolding.

contain a protease inhibitor cocktail (Roche). Cells were lysed using a microfluidizer (Microfluidics), followed by removal of cell debris by centrifugation at 39,000 x *g* at 4 °C for 30 minutes. His-tagged σNS was loaded onto a Ni-nitrilotriacetic acid (NTA) agarose column (Qiagen) and eluted using a gradient of 50 mM Tris-HCl (pH 8.0), 300 mM NaCl, and 250 mM imidazole. The eluted His-tagged σNS was concentrated

using a 10-kDa centrifugal filter unit (Millipore), dialyzed into 50 mM Tris-HCl (pH 8.0), 300 mM NaCl, and 10 mM imidazole, and incubated with TEV protease at 4 °C overnight. The cleaved protein mixture was reloaded onto a Ni-NTA agarose column to remove the His-TEV protease and uncleaved His-σNS. Cleaved σNS was subjected to ion-exchange chromatography using a Q-column (GE Healthcare) and

eluted with 20 mM Tris-HCl (pH 8.0) and 1 M NaCl. Eluted σNS was separated using an S200 16/60-increase column in 10 mM HEPES (pH 8.0) and 150 mM NaCl.

### Crystallization, data processing, structure determination, and refinement
Freshly purified Se-Met σNS-R6A, native σNS-R6A, and σNS-ΔN17 were used for initial crystallization screens immediately following size-exclusion chromatography. In each case, fractions with σNS proteins were collected and concentrated to -10 mg/mL in buffer containing 10 mM HEPES (pH 8.0) and 150 mM NaCl for crystallization. Crystals were propagated at 20 °C by hanging-drop vapor diffusion using a Mosquito crystallization robot (TTP LabTech) and imaged using a Rock Imager (Formulatrix). Each drop contained 0.2 μL of the σNS protein and 0.2 μL of crystallization buffer. Se-Met σNS-R6A produced crystals in the condition containing 0.1 M MES monohydrate (pH 6.5) and 1.6 M magnesium sulfate heptahydrate (Hampton Research), whereas native σNS-R6A and σNS-ΔN17 crystallized in the condition with 1.2% cholic acid derivatives mix, 0.1 M Buffer System 3 (pH 8.5), and 30% Precipitant Mix 3 (Morpheus III, Molecular Dimensions). Crystals were transferred into a cryoprotectant solution with 20% glycerol and flash-frozen in liquid nitrogen. X-ray diffraction data for σNS were collected using beamline 5.0.1 at the Advanced Light Source at the Lawrence Berkeley National Laboratory. Diffraction data were processed using the CCP4 software suite[31]. For Se-Met σNS-R6A, phases, and initial electron density maps were calculated by SAD phasing using SHELX[32], and automated model building with experimental phases was conducted using ARP/wARP[33] before iterative cycles of manual model building and refinement using COOT[34] and PHENIX[35]. Further model building was accomplished using COOT based on the difference maps[34] (see validation report for PDB ID: 8TKA). The structures of native σNS-R6A and σNS-ΔN17 were determined using molecular replacement with PHENIX[35]. The bile acid binding in these structures was validated by omit maps (see validation reports for PDB ID: 8TL8 and 8TL1). Data collection and refinement statistics following the final refinement cycle are provided in Table 1. Interactions between σNS and colic acid derivatives were analyzed using LigPlot+ v.2.1[36]. Figures were prepared using Chimera[37].

### Molecular dynamics simulation
Molecular dynamics simulation studies were conducted using the Desmond V 7.4 package (Schrodinger 2023.2). WT σNS dimers, with and without RNA (see Fig. S1a), and dimers of Se-Met σNS-R6A and σNS-R6A linked by domain-swapping N-terminal arm interactions (see Fig. S1b) were prepared using the Maestro protein preparation wizard. The dimers were solvated using the SPC model of the Desmond system builder and neutralized using $Na^+$ and $Cl^-$. Simulations were conducted using an OPLS−2005 force field for 10 ns at a temperature of 300°K and a pressure of 1 atm. The flexibility of the N-terminal arm of WT σNS dimers, with and without RNA, was analyzed using root mean square fluctuation (RMSF). Structural stability and binding affinity of the domain-swapped Se-Met σNS-R6A and σNS-R6A dimers were analyzed using RMSF and energy calculations, respectively.

### Preparation of RNA helix substrates
The RNA helix substrate was prepared by annealing two complementary RNA strands. One strand (RNA1) was labeled with HEX at the 5′ end, while the other was unlabeled (RNA2). Both strands were mixed at a 1:1 ratio in a 20-μL reaction mixture containing 25 mM HEPES-KOH (pH 8.0) and 25 mM NaCl. The mixture was incubated at 95 °C for 5 min and gradually cooled to 25 °C to produce duplex helices. HEX-labeled and unlabeled RNA strands were purchased from Integrated DNA Technologies USA. Oligonucleotides used in this study are shown in Table S1.

### RNA helix-destabilizing assay
RNA helix-destabilizing assays were conducted as described[38] with modifications. Briefly, various concentrations of σNS, σNS-R6A, and σNS-ΔN17 (2.5, 5, 10, and 20 μM) and 0.5 μM dsRNA helix substrate were incubated at 37 °C for 1 h in a reaction volume of 20 μL containing 25 mM HEPES-KOH (pH 8.0), 100 mM NaCl, 2 mM $MgCl_2$, 2 mM DTT, and 5 U RNase-OUT (Invitrogen). Reactions were terminated by the addition of 5 U proteinase K (NEB) for 15 min and 2.5 μL 5× loading buffer (100 mM Tris-HCl (pH 7.5), 50% glycerol, and bromophenol blue). The samples were electrophoresed in 4–20% native polyacrylamide gels (Genscript USA), and gels were scanned using a Typhoon 9200 imager (GE Healthcare). Bacterial-expressed maltose binding protein (MBP; 20 μM) and norovirus p41 protein (20 μM) were used as negative and positive controls, respectively. The effect of bile acids on σNS RNA helix-destabilizing activity was determined by incubating 20 and 40 μM of different components of bile acid derivatives, including TASAH, SGAH, CHAPS, and CHAPSO, with 20 μM σNS for 1 h before initiation of the helix-destabilization assay. Experiments were conducted three times.

### RNA strand-hybridization assay
The RNA strand-hybridization assay was conducted as described[38] with modifications. Various concentrations of σNS (2.5, 5, 10, and 20 μM) were incubated with two complementary stem-loop-structured RNA strands, one of which was HEX-labeled (RNA1) and the other was not (RNA3) (0.5 μM for each strand) at 37 °C for 1 h in a reaction volume of 20 μL containing 50 mM HEPES-KOH (pH 8.0), 2.5 mM $MgCl_2$, 2 mM DTT, 0.01% BSA, and 5 U RNase-OUT (Table S1). Reactions were terminated as described for RNA helix-destabilizing assays. Reaction products were resolved in 4–20% native polyacrylamide gels and scanned using a Typhoon 9200 imager. The two stem-loop RNA strands are shown in Fig. 6b. The stem-loop structures were predicted by RNAstructure[39]. The effect of bile acids on σNS RNA strand-hybridization activity was determined by incubating 20 and 40 μM of different components of bile acid derivatives, including TASAH, SGAH, CHAPS, and CHAPSO, with 20 μM σNS for 1 h before initiation of the RNA strand-hybridization assay. Experiments were conducted three times. Each band intensity was measured relative to the band intensity of the corresponding buffer and the quantification was done by ImageJ software[40].

### Reporting summary
Further information on research design is available in the Nature Portfolio Reporting Summary linked to this article.

## Data availability
Atomic coordinates and structure factors for the three crystal structures described in the manuscript have been deposited in the Protein Data Bank with accession codes, 8TKA for Se-Met σNS-R6A structure, 8TL8 for native σNS-R6A-bile acid structure, and 8TL1 for σNS-ΔN17-bile acid structure. The authors declare that all other data supporting the findings of this study are available within the body of the manuscript or the supplementary information file. Source data are provided with this paper.

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

## Acknowledgements

We are grateful to members of the Dermody and Prasad laboratories for many useful discussions. This work was supported by the U.S. Public Health Service award R01 AI032539, the Heinz Endowments (T.S.D.), and grant Q1279 from the Robert Welch Foundation (B.V.V.P). We acknowledge the Advanced Light Source (8.2.2) (Berkeley, CA) for X-ray data collection. The ALS-ENABLE beamlines are supported in part by U.S. Public Health Service award P30 GM124169. The Advanced Light Source is a Department of Energy Office of Science User Facility under Contract No. DE-AC02-05CH11231. The funders had no role in study design, data collection and analysis, decision to publish, or preparation of the manuscript.

## Author contributions

B.Z. and L.H. conceived, designed, and conducted experiments, analyzed data, contributed materials and analytic tools, and drafted the paper. S.K., N.N., C.H.L., X.S., and G.M.T. conceived and designed experiments, analyzed data, and contributed materials and analytic tools. B.S. assisted with the collection of X-ray diffraction data for

crystals analyzed at the ALS synchrotron facility. T.S.D. and B.V.V.P. conceived and designed experiments, analyzed the data, and drafted the paper. All authors reviewed, critiqued, and provided comments on the manuscript.

## Competing interests

The authors declare no competing interests.
