## [Peer Review File · Nature Communications]

Structure of Orthoreovirus RNA Chaperone σ NS, a Component of Viral Replication FactoriesReviewers' Comments:

Reviewer #1:

Remarks to the Author:

Zhou et al have determined the structure of orthoreovirus σ NS by x-ray crystallography and report that structure here. The native protein, which forms a stable dimer, assembles with RNA into helical filaments, but these filaments are not regular enough for high-resolution structure determination. For crystallization, the authors needed to use a previously characterized Arg-to-Ala mutant at position 6. The SeMet-derivitized protein then crystallized in space group P65, in effect making continuing helical rods running along the c direction in the crystal. In the helical assembly, an otherwise flexible, N-terminal arm inserts into a hydrophobic groove in a helically related subunit. In the presence of hydrophobic molecules (discovered presumably by serendipity when using various "additives" in crystallization), the native (non-SeMet) R6A mutant crystallized in a different space group, and the N-terminal 17 residues or so were disordered. Truncation of those 17 residues also eliminates the helical assembly. The central channel of the helical rods is positively charged, presumably illustrating the location of RNA when the protein includes it. If presented with dsRNA having a suitable overhang at the end, co-assembly appears to drive strand separation.

The structures are well determined, and the description of their features will be a valuable contribution to the dsRNA virus literature. It therefore merits publication in Nature Communications. The MS can benefit substantially from editorial attention, as outlined below. It is repetitious at many points, and seems to circumvent some issues at others. The narrative has an odd tendency to circle around the key point of a paragraph, rather than stating it straightforwardly, making the logic hard to follow in places. The Abstract is a good summary of what matters, and the items omitted from it, but described (in often distracting detail) in the text, should be less elaborated in the text than they are.

Line 38

See comment for line 255. Isn't the right thing to say: "for delivering the +-strand RNA for segment-selective incorporation into a replication-competent core particle"?

Section that begins on line 100 and then lines later in the text

The Results are clear, but later in the MS, the authors call the N-terminal arm "RNA binding". There's no evidence that the arm interacts with RNA -- it facilitates RNA binding by promoting oligomerization. The oligomer, with a positively charged channel, is almost certainly needed to bind RNA, so the effect of R6A is indirect. The authors seem confused on this key point.

Section that begins on line 121

Call the section "Crystal structure of σ NS-R6A"

Delete lines 122-142 and substitute: "We determined the structure of the σ NS-R6A mutant by SeMet SAD phasing to a resolution of 3.0Å (Table 1). The crystals were in space group P65, with two molecules in the asymmetric unit. The dimeric asymmetric unit forms a helical assembly that extends continuously from one unit cell to the next along the c-axis of the crystal, with a non-crystallographic axis relating the two molecule, perpendicular to the helix axis. Fig. 2a shows the folded structure of a subunit and Fig. 2b, the arrangement of secondary structures." [What angle does the dyad make with a, so that the crystals are P65 and not P6522? Or is the dyad not quite perpendicular to the 6-fold screw? If it incorporates ssRNA, is need to do so with some polarity, so there's something to discuss here that is not in the Discussion. Leave out the old-fashioned recitation of secondary structural elements, etc. -- a good figure is enough.]

Lines 137ff.

The MS simulation is not the right way to show flexibility when there are data that show the same thing. Delete it, as it just distract and draws attention away from real data. They have crystals in the presence of bile acids that show that the N-terminal arm is disordered. That's a solid, experimental result.

Section that begins line 144.

Incorporate this AND the following section with the previous one into a SINGLE section. The current title is inappropriate anyhow. Only data for a stable dimer in solution can support the current title. Structures do not by themselves report "stability", although often one can make inferences from them. There are already data in line 114 showing dimerization in solution. Moreover, I think that the authors don't fully appreciate that docking of flexible arms in the most common way that viral assemblies are formed. Consider the NPs of negative strand RNA viruses, the network of arms in non-enveloped, positive-strand RNA viruses, etc. So their emphasis on that utterly unsurprising aspect of the structure is a bit misplaced.

The second paragraph of the unified structure description should incorporate the essence of the line 144 et seq section, as follows: "Fig. 3a shows the structure of the dimer. The N-terminal arms of the two subunits project laterally, away from the body of the subunit, and insert into grooves in the neighboring dimers, stabilizing the helical assembly. The details of the dimer interface are in Figs. 3b and c. Two other structures, described below, confirm the choice of which twofold axis represents the dimeric building block of the helix." [Leave the obsessive description of residues and contacts at the interface to the figure -- no one wants to read through all those residue names.]

The third paragraph of the unified structure section should incorporate the essence of the section starting on line 157, as follows: "Insertion of the projecting N-terminal arms of the subunit into grooves in the neighboring subunits defines, together with extended dimer-dimer interfaces, the helical assembly of σ NS-R6A dimers (Fig. 4a-d). The assembly has a central channel, ~ 40 Å in diameter ; its outside diameter is about 150 Å. These dimensions are consistent with the assemblies of σ NS with RNA seen by cryo-EM. Native σ NS, not substituted with Se-Met, does not form such arrays. Preferential interactions of the Se atom (replacing S) with residues lining the groove into which it inserts might account for the difference.

Line 180.

Delete "remarkably". There's nothing remarkable about the observation. Indeed, exactly what you'd expect if there isn't a continuous helical assembly in the crystals.

Line 184. Delete "strikingly".

How do you "strikingly occupy" anything. In any case, get rid of emotion or personal reactions from scientific papers. Use of "remarkably", "strikingly", "interestingly", etc., is just a lazy way to avoid writing sentences that convey what's "striking" or "interesting" about whatever is being described.

Line 193

Get rid of "interestingly".

Line 199

Delete "collectively". Each crystal structure provides good evidence that dimerization accounts for the size-exclusion results.

Line 200

(i) WT σ Ns forms dimers (dimerization isn't a "tendency", and "has a tendency to form" is simply a weasel phrase).

Lines 205

Only cholesterol-like? Did they try other hydrophobic molecules? Detergents like DDM? etc.

Line 208

"exhibits" -- what's wrong with "has"? (likewise for "displays" in line 225). Why do Latinic polysyllabics impress people as decorating their prose, when the monosyllabic, Anglo-Saxon derived equivalents are

cleaner, simpler, and better expository style?

Lines 224

"catalyzes" -- if σNs is a catalyst, something must re-set it after it delivers RNA, and that's where the input of free energy would come in to make it a helicase

Line 232

Does a helicase need to split ATP? Obviously, it needs to do so in order to recycle, but not for a one-way process. In other words, the distinction between "helicase" and "chaperones" seems to this reviewer too fuzzy for the line 232 statement to be useful. (See next comment)

Line 255ff

This sentence may need rewriting, as it confused me at first until I consulted Dermody's chapter in the latest edition of Fields' Virology. I think the word that needs changing is "loading" -- perhaps to "association with" -- as on overly quick reading, the sentence seemed to imply that the polymerase started working on the RNA before assembly (that was a false impression on my part, but I got hung up on it until careful re-reading).

Line 269

What is a higher-order NSP2 octamer? I thought NSP2 was an octamer, neither higher- or lower-order.

Section beginning line 277

This section seems much too long. It is mainly just a riff on what has gone before. Flexible arms that dock (a verb I prefer to "domain-swap" for what is not really a domain, but that's a matter of taste) onto the neighbor in an assembly are a dime a dozen (see above). And there's no reason to suppose that the arm binds RNA -- it may simply stabilize the assembly to retain RNA. So this reviewer finds that the speculations in the second paragraph go a bit beyond what he believes should be the "license to speculate" that indeed one earns by providing solid results. at least acknowledge that the RNA could associate with the much larger basic patch that winds up in the tunnel and that the arm docking allows the polymerization that would stabilize everything.

Section beginning line 306

Bile acid binding doesn't "tamper" with anything -- much less with the arm. It prevents the groove, presumably, from accepting the arm, but it doesn't interact with the arm directly.

First sentence: "Bile acid derivatives bind to the same site" (The reader won't care what was serendipitous and what fascinates the authors -- what matters is to fascinate the reader.)

This reviewer finds the "entropy transfer" business a bit unfounded, but if the authors want to invoke that notion, they are probably entitled to do so in a discussion. Just warning: it turned off this reader. I also don't believe that RNA interacts with the arm (see above) but rather than it nucleates assembly by interacting with the much more substantial basic patch that faces the interior of the tube.

Note: the authors should open the Discussion (or some very early paragraph) by justifying taking the helical assembly in the crystal as representing (approximately) the RNA-incorporating filaments seen by EM. They present the key observation in a sentence (about diameter) that I rewrote above. But it is important to make it clear that their interpretation relies on the similarity of the crystallographic helices with those seen in the EM of RNP filaments. In other words, they need to communicate to non-structural readers that what they've seen in the crystal is a more ordered (because of packing) version of what they are confident is happening in solution and in cells.

Reviewer #2:

Remarks to the Author:

The manuscript by Zhao, Hu et al provides profound insights into the replication mechanism of

mammalian orthoreoviruses, particularly the role of σ NS in unwinding viral RNA and its RNA chaperone activity. This work seems to be building upon previously identified roles of σ NS and provides a more in-depth mechanistic understanding. The study elucidates the role of mammalian orthoreovirus σ NS in forming filamentous assemblies via dimeric units through domain-swapping interactions of the N-terminal arms. They demonstrate that just like avian reovirus σ NS, the mammalian RV σ NS exhibits RNA chaperone activity without the need for a metal ion or ATP, making it a genuine RNA chaperone. Serendipitously, the team uncovers that bile acid derivatives can bind to the σ NS at the same site as its N-terminal arm and shows that this compound may interfere with the RNA chaperone activity of σ NS. This is of notable importance to the broader field of virology. While the structural function of σ NS has been elaborated upon in this manuscript, comparisons with proteins like rotavirus NSP2 and rice black-streaked dwarf virus (RBSDV) P9-1 offer a broader perspective, situating the findings within established literature. The study is methodologically sound and it employs a combination of crystallographic and biochemical methods, and addresses a long-standing issue of not being able to obtain an atomic structure of this important RNA chaperone.

The authors also propose that the chaperone activity of the protein assists RNA replication – I suggest they add some references to support this model or provide further experimental evidence of the role of σ NS in facilitating replication. A good example of in-depth characterisation of the role of NSP2 in rotavirus replication was shown by Vende et al, *Virology*, 2003, in which the authors show that NSP2 inhibited the synthesis of dsRNA from viral mRNA *in vitro*, in a concentration-dependent manner. The inhibition was overcome by adding increasing amounts of viral mRNA or nonviral ssRNA to the system, indicating that the inhibition was mediated by the nonspecific RNA-binding activity of NSP2, therefore, it is also possible that σ NS binding is not essential for presenting the RNA to the polymerase, but rather is involved in promoting RNA assembly. The data analysis and interpretation are generally robust.

While the manuscript offers compelling evidence linking the structure and function of σ NS to its role in viral replication, it might benefit from further in-depth biochemical or biophysical experiments to solidify some of the proposed mechanisms. Given that there are no available structures of any σ NS protein, it would be interesting to add whether AlphaFold models were consistent with the new structure.

Overall, this manuscript presents valuable insights into the function of σ NS in viral replication, enriching our understanding of mammalian orthoreovirus replication. While the results are significant and add to the established literature, a few improvements can be made to bolster the claims further and make the manuscript more accessible to a broader audience. With the suggested revisions, the manuscript would be a strong contribution to the field.

Additional points for consideration:

1. The use of selenomethionine (Se-Met) in the σ NS-R6A structure, where selenium replaces sulfur, raises interesting questions about how this substitution may influence the behavior of the protein, particularly in the context of the flexibility of the N-terminal arms. Given that the manuscript did not delve deeply into the specific effects of the selenium substitution, it would be beneficial to clarify if the observed characteristics are truly reflective of the protein's natural behavior or influenced by the Se-Met substitution (or have something to do with crystal packing). Perhaps, the authors could clarify this by running additional MD simulations as they did in their paper.
2. Lines 160-170: The claim that the helical assembly in crystal corresponds to the filaments previously seen by cryo-EM should be further substantiated. Crystal packing can influence the helical symmetry, and it is possible that it differs in solution.
3. Lines 294-304: The authors imply that Arg6 is directly involved in RNA binding and effectively competes with N-terminal tail swapping and higher oligomer formation. Given that higher oligomers are necessary for RNA binding this is somewhat contradictory. Perhaps Arg6 is engaged in an important interaction with the recipient subunit during domain swapping, and its mutation to alanine affects oligomer formation and consequently impairs RNA binding.
4. The observation that a small molecule ligand/bile acid derivative may be competing with the N-terminal arm is fascinating. It would be great if the authors could further elaborate by providing some additional quantitation of ligand's binding to σ NS, e.g., is it nanomolar/micromolar affinity? It would be interesting to understand whether this direction could be taken further for potentially developing new

antivirals.

5. Figures and visual representation:

a. It would be beneficial to have 260 and 280 nm absorbances in Figure 1a to help readers see the contribution of nucleic acid binding by σ NS WT and R6A (and to see whether they have similar purities with respect to nucleic acids). The Figure legend should also include the information about the SEC column used for panels a and b.

b. In Line 140 the authors state that the addition of RNA reduces the RMSF of the N-terminal tail by 25% - perhaps, they could add these data in their Fig.2c as I couldn't find the reference to this result.

Reviewer #3:

Remarks to the Author:

Reovirus is a 10-segmented double-stranded RNA virus that replicates in the cytoplasm of the host cell. A characteristic feature of reovirus infection is the formation of viral factories (VFs), which are biomolecular condensates that serve as concentrated sites for RNA genome segment assortment/packaging into nascent particles and intra-particle genome replication. The reovirus non-structural protein σ NS is required for VF nucleation (along with its binding partner μ NS), and it may also play direct roles as an RNA chaperone to facilitate segment assortment/packaging and replication.

Previous work from the authors has shown σ NS forms heterogeneous higher-order oligomers as well as filamentous structures when bound to RNA. An R6A σ NS mutant was found to be defective in both RNA binding and higher-ordered oligomer (i.e., filament) formation; these properties made the mutant more structurally tractable as compared to the WT σ NS protein. In this current study, the authors elucidated the X-ray crystal structure of the R6A σ NS mutant, showing that it forms a stable dimer. Interestingly, the selenomethionine (Se-Met)-substituted version of R6A σ NS was able to form an octamer. While the N-terminal arm of σ NS was disordered in the non-substituted (native) R6A σ NS, it was resolved in the Se-Met-substituted structure and revealed to participate in domain-swapping interactions that facilitate filament formation. Bile acid derivatives were included during the crystallization of native R6A σ NS, and they were found to occupy the same position as the N-terminal arm, thereby providing an explanation for the disruption of filaments. This idea was further supported using a deletion mutant that lacked the N-terminal arm (Δ N17). Biochemical experiments examined the N-terminal mutants for dsRNA duplex destabilizing activity as compared to WT σ NS, and the inclusion of bile acids in the reaction was found to diminish this activity for WT σ NS. These results suggest that σ NS filament formation, mediated by the N-terminal arm, is important for the RNA chaperone activity of this protein.

This paper is generally well-written and the new structures of σ NS will be of interest to the virology community. However, I do have some concerns about the rigor and reproducibility of the biochemical RNA destabilizing/annealing activities shown in Fig. 6 (see comment 5 below). The methods section is also missing some experimental details, making it difficult to assess some of the data (see specific comments below). Additional editorial suggestions are listed below for the authors' consideration.

Specific comments:

1. Lines 36-37: Please restate "we discovered that σ NS displays RNA helix destabilizing and annealing activities." These activities were already reported by Borodavka et al. in 2015 (ref. 16) for avian reovirus; this study confirms that the mammalian orthoreovirus σ NS protein also performs these functions.
2. Line 37-38: Please remove/temper the statement "...likely essential for presenting mRNA to the viral RNA-dependent RNA polymerase for genome replication." No evidence is provided for the role of RNA destabilizing during viral replication.
3. Line 50 and throughout: Please change "Reoviridae" to "Spinoreoviridae" per the new ICTV

designation (PMID: 36394457).

4. Line 81: Define "ssRNA" as "single-stranded RNA".

5. Fig. 6, Results lines 216-232, and Methods lines 380-399: There is a lack of information regarding the details of the RNA destabilizing and RNA annealing experiments, raising both confusion as well as concerns about rigor. What was the size and source of the RNAs used in the reactions? Are these the same RNA substrates used by Yang et al., (ref 35)? Adding molecular weight markers would increase rigor of gels. How much of the control proteins (MBP and NV p41) were used in reactions and what was the source of these proteins? How many times were the reactions completed? Were the results quantified? Why does "boiling two ssRNA molecules promote strand annealing" as stated in the legend for 6h. Did the authors test the N-terminal mutants for RNA annealing? If not, why not? Why wasn't the effect of bile tested in the context of RNA annealing too?

6. Lines 254-257: The statement "These findings implicate σ NS in unwinding viral RNA for loading...virion assembly" is overstated given the available data. Consider removing this sentence here and waiting to speculate in the Discussion section.

7. Lines 267-268: Please add "under crystallographic conditions" to the sentence describing NSP2 inter-octamer association. To my knowledge, there is no evidence that NSP2 performs these inter-octamer interactions in solution or in cells.

8. Lines 307-308: It would be interesting to add purified N-terminal arm domain only to the in vitro reactions.

9. Line 321: Please restate "our discovery" given that this activity has already been described, at least for avian σ NS.

10. Line 335: Please add the strain of reovirus used for σ NS proteins (ie., T3D?).

11. Line 340: The methods seem to indicate that all protein expressions were done with 10 mg/mL Se-Met. Please rephrase.

12. Lines 345-346: Please add information about how the A280 was measured for the elution profiles in Fig. 1.

13. Please add information about protein purification yield, SDS-PAGE analysis (if performed) and protein storage conditions.

14. There are no methods for the molecular dynamics (Fig. 2c) or the molecular docking experiments (Fig. 6d-e). Please add.

Nature Communications manuscript NCOMMS-23-39686

We thank all the reviewers for their positive comments and suggestions. We have addressed each reviewer's comments in detail and revised the text accordingly. Our responses are in blue. Changes are highlighted in the 'marked' text.

REVIEWER COMMENTS

Reviewer #1 (Remarks to the Author):

Zhou et al have determined the structure of orthoreovirus σ NS by x-ray crystallography and report that structure here. The native protein, which forms a stable dimer, assembles with RNA into helical filaments, but these filaments are not regular enough for high-resolution structure determination. For crystallization, the authors needed to use a previously characterized Arg-to-Ala mutant at position 6. The SeMet-derivitized protein then crystallized in space group P65, in effect making continuing helical rods running along the c direction in the crystal. In the helical assembly, an otherwise flexible, N-terminal arm inserts into a hydrophobic groove in a helically related subunit. In the presence of hydrophobic molecules (discovered presumably by serendipity when using various "additives" in crystallization), the native (non-SeMet) R6A mutant crystallized in a different space group, and the N-terminal 17 residues or so were disordered. Truncation of those 17 residues also eliminates the helical assembly. The central channel of the helical rods is positively charged, presumably illustrating the location of RNA when the protein includes it. If presented with dsRNA having a suitable overhang at the end, co-assembly appears to drive strand separation.

The structures are well determined, and the description of their features will be a valuable contribution to the dsRNA virus literature. It therefore merits publication in Nature Communications. The MS can benefit substantially from editorial attention, as outlined below. It is repetitious at many points and seems to circumvent some issues at others. The narrative has an odd tendency to circle around the key point of a paragraph, rather than stating it straightforwardly, making the logic hard to follow in places. The Abstract is a good summary of what matters, and the items omitted from it, but described (in often distracting detail) in the text, should be less elaborated in the text than they are.

We thank the reviewer for the positive and constructive comments.

Line 38

See comment for line 255. Isn't the right thing to say: "for delivering the +-strand RNA for segment-selective incorporation into a replication-competent core particle"?

Section that begins on line 100 and then lines later in the text

The Results are clear, but later in the MS, the authors call the N-terminal arm "RNA binding". There's no evidence that the arm interacts with RNA -- it facilitates RNA binding by promoting oligomerization. The oligomer, with a positively charged channel, is almost

certainly needed to bind RNA, so the effect of R6A is indirect. The authors seem confused on this key point.

In the section that begins at line 100, we describe the results of previous experiments that show that the σ NS-R6A mutant lacks RNA-binding activity. We are cautious in suggesting the involvement of the N-terminal arm in RNA binding. In response to this comment and a similar comment by Reviewer 2, we revised the sentences (lines 282-287) in the Discussion to indicate that Arg6 “influences” RNA-binding capacity by either directly binding to RNA or disrupting σ NS oligomerization. The following sentences continue with the idea that RNA-binding capacity is linked to oligomerization.

Section that begins on line 121

Call the section “Crystal structure of σ NS-R6A”

Delete lines 122-142 and substitute: “We determined the structure of the σ NS-R6A mutant by SeMet SAD phasing to a resolution of 3.0Å (Table 1). The crystals were in space group P65, with two molecules in the asymmetric unit. The dimeric asymmetric unit forms a helical assembly that extends continuously from one unit cell to the next along the c-axis of the crystal, with a non-crystallographic axis relating the two molecules, perpendicular to the helix axis. Fig. 2a shows the folded structure of a subunit and Fig. 2b, the arrangement of secondary structures.” [What angle does the dyad make with a, so that the crystals are P65 and not P6522? Or is the dyad not quite perpendicular to the 6-fold screw? If it incorporates ssRNA, is needed to do so with some polarity, so there’s something to discuss here that is not in the Discussion. Leave out the old-fashioned recitation of secondary structural elements, etc. -- a good figure is enough.]

We agree and have revised this section (lines 119-152), as suggested.

The MS simulation is not the right way to show flexibility when there are data that show the same thing. Delete it, as it just distracts and draws attention away from real data. They have crystals in the presence of bile acids that show that the N-terminal arm is disordered. That’s a solid, experimental result.

Based on the comments of Reviewer 2, we retained this section, but we placed the figure in the supplementary material section (Figure S2).

Section that begins line 144.

Incorporate this AND the following section with the previous one into a SINGLE section. The current title is inappropriate anyhow. Only data for a stable dimer in solution can support the current title. Structures do not by themselves report “stability,” although often one can make inferences from them. There are already data in line 114 showing dimerization in solution. Moreover, I think that the authors don’t fully appreciate that docking of flexible arms in the most common way that viral assemblies are formed. Consider the NPs of negative strand RNA viruses, the network of arms in non-enveloped, positive-strand RNA viruses, etc. So, their emphasis on that utterly unsurprising aspect of

the structure is a bit misplaced.

We modified this section according to the suggestions of the reviewer (lines 119-152).

The second paragraph of the unified structure description should incorporate the essence of the line 144 et seq section, as follows: “Fig. 3a shows the structure of the dimer. The N-terminal arms of the two subunits project laterally, away from the body of the subunit, and insert into grooves in the neighboring dimers, stabilizing the helical assembly. The details of the dimer interface are in Figs. 3b and c. Two other structures, described below, confirm the choice of which twofold axis represents the dimeric building block of the helix.” [Leave the obsessive description of residues and contacts at the interface to the figure -- no one wants to read through all those residue names.]

The third paragraph of the unified structure section should incorporate the essence of the section starting on line 157, as follows: “Insertion of the projecting N-terminal arms of the subunit into grooves in the neighboring subunits defines, together with extended dimer-dimer interfaces, the helical assembly of σ NS-R6A dimers (Fig. 4a-d). The assembly has a central channel, ~ 40 Å in diameter; its outside diameter is about 150 Å. These dimensions are consistent with the assemblies of σ NS with RNA seen by cryo-EM. Native σ NS, not substituted with Se-Met, does not form such arrays. Preferential interactions of the Se atom (replacing S) with residues lining the groove into which it inserts might account for the difference.

We modified these sections according to the suggestions of the reviewer (lines 119-152).

Line 180.

Delete “remarkably.” There’s nothing remarkable about the observation. Indeed, exactly what you’d expect if there isn’t a continuous helical assembly in the crystals.

Line 184. Delete “strikingly.”

How do you “strikingly occupy” anything. In any case, get rid of emotion or personal reactions from scientific papers. Use of “remarkably,” “strikingly,” “interestingly,” etc., is just a lazy way to avoid writing sentences that convey what’s “striking” or “interesting” about whatever is being described.

Line 193

Get rid of “interestingly.”

Line 199

Delete “collectively.” Each crystal structure provides good evidence that dimerization accounts for the size-exclusion results.

Line 200

(i) WT σ Ns forms dimers (dimerization isn’t a “tendency,” and “has a tendency to form” is simply a weasel phrase).

We agree and removed “remarkably,” “strikingly,” “interestingly,” “collectively,” and “tendency.”

Lines 205

Only cholesterol-like? Did they try other hydrophobic molecules? Detergents like DDM? etc.

We did not try other detergents in these experiments.

Line 208

“exhibits” -- what’s wrong with “has”? (likewise, for “displays” in line 225). Why do Latinic polysyllabics impress people as decorating their prose, when the monosyllabic, Anglo-Saxon derived equivalents are cleaner, simpler, and better expository style?

We replaced “exhibits” and “displays” with “has.”

Lines 224

“catalyzes” -- if σ NS is a catalyst, something must re-set it after it delivers RNA, and that’s where the input of free energy would come in to make it a helicase

We replaced “catalyzes” with “facilitates.” We clarified elsewhere in the text that σ NS is not a helicase, as the σ NS helix-destabilizing and RNA chaperone activities do not require ATP hydrolysis.

Line 232

Does a helicase need to split ATP? Obviously, it needs to do so in order to recycle, but not for a one-way process. In other words, the distinction between “helicase” and “chaperones” seems to this reviewer too fuzzy for the line 232 statement to be useful. (See next comment)

We clarified that the σ NS helix-destabilizing and RNA chaperone activities do not require ATP hydrolysis (lines 216-218).

Line 255ff

This sentence may need rewriting, as it confused me at first until I consulted Dermody’s chapter in the latest edition of Fields’ Virology. I think the word that needs changing is “loading” -- perhaps to “association with” -- as on overly quick reading, the sentence seemed to imply that the polymerase started working on the RNA before assembly (that was a false impression on my part, but I got hung up on it until careful re-reading).

We changed “loading” to “delivered to” (line 245).

Line 269

What is a higher-order NSP2 octamer? I thought NSP2 was an octamer, neither higher- or

lower-order.

We concur and revised the text to “inter-octameric association” (line 259).

Section beginning line 277

This section seems much too long. It is mainly just a riff on what has gone before. Flexible arms that dock (a verb I prefer to “domain-swap” for what is not really a domain, but that’s a matter of taste) onto the neighbor in an assembly are a dime a dozen (see above). And there’s no reason to suppose that the arm binds RNA -- it may simply stabilize the assembly to retain RNA. So, this reviewer finds that the speculations in the second paragraph go a bit beyond what he believes should be the “license to speculate” that indeed one earns by providing solid results. at least acknowledge that the RNA could associate with the much larger basic patch that winds up in the tunnel and that the arm docking allows the polymerization that would stabilize everything.

See our response to the comment above about Arg6. We do not know whether the N-terminal arm of σ NS binds to RNA directly. Therefore, we reworded the sentences in this section (lines 282-287).

Section beginning line 306

Bile acid binding doesn’t “tamper” with anything -- much less with the arm. It prevents the groove, presumably, from accepting the arm, but it doesn’t interact with the arm directly. First sentence: “Bile acid derivatives bind to the same site” (The reader won’t care what was serendipitous and what fascinates the authors -- what matters is to fascinate the reader.)

We concur and removed “serendipitous” from the sentence in question.

This reviewer finds the “entropy transfer” business a bit unfounded, but if the authors want to invoke that notion, they are probably entitled to do so in a discussion. Just warning: it turned off this reader.

The concept of entropy transfer is suggested for other viral RNA chaperones and is consistent with our observations about σ NS. We provided a reference for our speculation (line 282).

I also don’t believe that RNA interacts with the arm (see above) but rather than it nucleates assembly by interacting with the much more substantial basic patch that faces the interior of the tube.

We concur and clarified throughout the text that the σ NS arm may interact with RNA or nucleate assembly of σ NS oligomers (lines 282-287).

Note: the authors should open the Discussion (or some very early paragraph) by justifying taking the helical assembly in the crystal as representing (approximately) the RNA-

incorporating filaments seen by EM. They present the key observation in a sentence (about diameter) that I rewrote above. But it is important to make it clear that their interpretation relies on the similarity of the crystallographic helices with those seen in the EM of RNP filaments. In other words, they need to communicate to non-structural readers that what they've seen in the crystal is a more ordered (because of packing) version of what they are confident is happening in solution and in cells.

As suggested, we included this important point in the introductory paragraph of the Discussion section (lines 243-244).

Reviewer #2 (Remarks to the Author):

The manuscript by Zhao, Hu et al provides profound insights into the replication mechanism of mammalian orthoreoviruses, particularly the role of σ NS in unwinding viral RNA and its RNA chaperone activity. This work seems to be building upon previously identified roles of σ NS and provides a more in-depth mechanistic understanding. The study elucidates the role of mammalian orthoreovirus σ NS in forming filamentous assemblies via dimeric units through domain-swapping interactions of the N-terminal arms. They demonstrate that just like avian reovirus σ NS, the mammalian RV σ NS exhibits RNA chaperone activity without the need for a metal ion or ATP, making it a genuine RNA chaperone. Serendipitously, the team uncovers that bile acid derivatives can bind to the σ NS at the same site as its N-terminal arm and shows that this compound may interfere with the RNA chaperone activity of σ NS. This is of notable importance to the broader field of virology. While the structural function of σ NS has been elaborated upon in this manuscript, comparisons with proteins like rotavirus NSP2 and rice black-streaked dwarf virus (RBSDV) P9-1 offer a broader perspective, situating the findings within established literature. The study is methodologically sound, employs a combination of crystallographic and biochemical methods, and addresses a long-standing issue of not obtaining an atomic structure of this important RNA chaperone.

The authors also propose that the chaperone activity of the protein assists RNA replication – I suggest they add some references to support this model or provide further experimental evidence of the role of σ NS in facilitating replication. A good example of in-depth characterization of the role of NSP2 in rotavirus replication was shown by Vende et al, *Virology*, 2003, in which the authors show that NSP2 inhibited the synthesis of dsRNA from viral mRNA in vitro, in a concentration-dependent manner. The inhibition was overcome by adding increasing amounts of viral mRNA or nonviral ssRNA to the system, indicating that the nonspecific RNA-binding activity of NSP2 mediated the inhibition, therefore, it is also possible that σ NS binding is not essential for presenting the RNA to the polymerase, but rather is involved in promoting RNA assembly. The data analysis and interpretation are generally robust.

While the manuscript offers compelling evidence linking the structure and function of σ NS to its role in viral replication, it might benefit from further in-depth biochemical or biophysical experiments to solidify some of the proposed mechanisms. Given that there are no available structures of any σ NS protein, it would be interesting to add whether

AlphaFold models were consistent with the new structure.

Overall, this manuscript presents valuable insights into the function of σ NS in viral replication, enriching our understanding of mammalian orthoreovirus replication. While the results are significant and add to the established literature, a few improvements can be made to bolster the claims further and make the manuscript more accessible to a broader audience. With the suggested revisions, the manuscript would strongly contribute to the field.

We thank the reviewer for the positive comments. As suggested, we added the Vende et al., 2003 reference (line 324, ref 30). The model predicted by AlphaFold2 is consistent with the crystal structure. We do not think it is necessary to include that model in the main text.

Additional points for consideration:

1. The use of selenomethionine (Se-Met) in the σ NS-R6A structure, where selenium replaces sulfur, raises interesting questions about how this substitution may influence the behavior of the protein, particularly in the context of the flexibility of the N-terminal arms. Given that the manuscript did not delve deeply into the specific effects of the selenium substitution, it would be beneficial to clarify if the observed characteristics are truly reflective of the protein's natural behavior or influenced by the Se-Met substitution (or have something to do with crystal packing). Perhaps the authors could clarify this by running additional MD simulations as they did in their paper.

We conducted additional molecular dynamics simulations as suggested and incorporated these data into Figure S2.

2. Lines 160-170: The claim that the helical assembly in crystal corresponds to the filaments previously seen by cryo-EM should be further substantiated. Crystal packing can influence the helical symmetry, and it is possible that it differs in solution.

The symmetrical organization of the helical symmetry is a consequence of crystal packing. In the absence of a cryo-EM structure of the σ NS filaments, which is beyond the scope of this study, we only can suggest that they are similar based on their similar overall diameters. We revised the text reflecting this conclusion (lines 138-140).

3. Lines 294-304: The authors imply that Arg6 is directly involved in RNA binding and effectively competes with N-terminal tail swapping and higher oligomer formation. Given that higher oligomers are necessary for RNA binding this is somewhat contradictory. Perhaps Arg6 is engaged in an important interaction with the recipient subunit during domain swapping, and its mutation to alanine affects oligomer formation and consequently impairs RNA binding.

We revised a sentence in this section (lines 282-287) to state the possibility that Arg6 "influences" the RNA-binding capacity of σ NS by either directly binding to RNA or

disrupting σ NS oligomerization. The sentences that follow continue with the idea that RNA-binding capacity is linked to oligomerization.

4. The observation that a small molecule ligand/bile acid derivative may be competing with the N-terminal arm is fascinating. It would be great if the authors could further elaborate by providing some additional quantitation of ligand's binding to σ NS, e.g., is it nanomolar/micromolar affinity? It would be interesting to understand whether this direction could be taken further for potentially developing new antivirals.

We concur with the reviewer that this is an interesting idea. We plan to focus our future studies on quantification of σ NS-RNA binding affinity toward development of proof-of-concept antivirals.

5. Figures and visual representation:

a. It would be beneficial to have 260 and 280 nm absorbances in Figure 1a to help readers see the contribution of nucleic acid binding by σ NS WT and R6A (and to see whether they have similar purities with respect to nucleic acids). The Figure legend should also include the information about the SEC column used for panels a and b.

We included the requested data in the supplementary materials (Figure S1). We also included information about the SEC column used for Figure 1 panels a and b in the figure legend.

b. In Line 140 the authors state that the addition of RNA reduces the RMSF of the N-terminal tail by 25% - perhaps, they could add these data in their Fig.2c as I couldn't find the reference to this result.

We included this information in Figure S2 and Table S2 and added a paragraph about the MD calculations to the Methods section (lines 374-384).

Reviewer #3 (Remarks to the Author):

Reovirus is a 10-segmented double-stranded RNA virus that replicates in the cytoplasm of the host cell. A characteristic feature of reovirus infection is the formation of viral factories (VFs), which are biomolecular condensates that serve as concentrated sites for RNA genome segment assortment/packaging into nascent particles and intra-particle genome replication. The reovirus non-structural protein σ NS is required for VF nucleation (along with its binding partner μ NS), and it may also play direct roles as an RNA chaperone to facilitate segment assortment/packaging and replication.

Previous work from the authors has shown σ NS forms heterogeneous higher-order oligomers as well as filamentous structures when bound to RNA. An R6A σ NS mutant was found to be defective in both RNA binding and higher-ordered oligomer (i.e., filament) formation; these properties made the mutant more structurally tractable as compared to

the WT σ NS protein. In this current study, the authors elucidated the X-ray crystal structure of the R6A σ NS mutant, showing that it forms a stable dimer. Interestingly, the selenomethionine (Se-Met)-substituted version of R6A σ NS was able to form an octamer. While the N-terminal arm of σ NS was disordered in the non-substituted (native) R6A σ NS, it was resolved in the Se-Met-substituted structure and revealed to participate in domain-swapping interactions that facilitate filament formation. Bile acid derivatives were included during the crystallization of native R6A σ NS, and they were found to occupy the same position as the N-terminal arm, thereby providing an explanation for the disruption of filaments. This idea was further supported using a deletion mutant that lacked the N-terminal arm (Δ N17). Biochemical experiments examined the N-terminal mutants for dsRNA duplex destabilizing activity as compared to WT σ NS, and the inclusion of bile acids in the reaction was found to diminish this activity for WT σ NS. These results suggest that σ NS filament formation, mediated by the N-terminal arm, is important for the RNA chaperone activity of this protein.

This paper is generally well-written and the new structures of σ NS will be of interest to the virology community. However, I do have some concerns about the rigor and reproducibility of the biochemical RNA destabilizing/annealing activities shown in Fig. 6 (see comment 5 below). The methods section is also missing some experimental details, making it difficult to assess some of the data (see specific comments below). Additional editorial suggestions are listed below for the authors' consideration.

We thank the reviewer for the positive comments.

Specific comments:

1. Lines 36-37: Please restate "we discovered that σ NS displays RNA helix destabilizing and annealing activities." These activities were already reported by Borodavka et al. in 2015 (ref. 16) for avian reovirus; this study confirms that the mammalian orthoreovirus σ NS protein also performs these functions.

We added "mammalian" in the abstract (line 36) and referenced the studies on avian σ NS by Borodavka and colleagues in the text.

2. Line 37-38: Please remove/temper the statement "...likely essential for presenting mRNA to the viral RNA-dependent RNA polymerase for genome replication." No evidence is provided for the role of RNA destabilizing during viral replication.

We revised this sentence to read "...which may be required to present mRNA to the viral RNA-dependent RNA polymerase for genome replication." (Line 36)

3. Line 50 and throughout: Please change "Reoviridae" to "Spinoreoviridae" per the new ICTV designation (PMID: 36394457).

We changed "Reoviridae" to "Spinoreoviridae," as requested (lines 48 and 95, ref 4)

4. Line 81: Define “ssRNA” as “single-stranded RNA”.

We defined “ssRNA” as “single-stranded RNA,” as requested.

5. Fig. 6, Results lines 216-232, and Methods lines 380-399: There is a lack of information regarding the details of the RNA destabilizing and RNA annealing experiments, raising both confusion as well as concerns about rigor. What was the size and source of the RNAs used in the reactions? Are these the same RNA substrates used by Yang et al., (ref 35)? Adding molecular weight markers would increase rigor of gels. How much of the control proteins (MBP and NV p41) were used in reactions and what was the source of these proteins? How many times were the reactions completed? Were the results quantified? Why does “boiling two ssRNA molecules promote strand annealing” as stated in the legend for 6h. Did the authors test the N-terminal mutants for RNA annealing? If not, why not? Why wasn't the effect of bile tested in the context of RNA annealing too?

We added all of this information to the text and revised Figure 6. In addition, we have added Figure S3 and Table S1, which further address the comments about quantification.

6. Lines 254-257: The statement “These findings implicate σ NS in unwinding viral RNA for loading...virion assembly” is overstated given the available data. Consider removing this sentence here and waiting to speculate in the Discussion section.

We revised this sentence as follows: “These findings suggest that σ NS unwinds the viral RNA for delivery to the viral RNA-dependent RNA polymerase during genome replication and promotes interactions between the viral RNAs to facilitate their selective encapsidation during virion assembly.”

7. Lines 267-268: Please add “under crystallographic conditions” to the sentence describing NSP2 inter-octamer association. To my knowledge, there is no evidence that NSP2 performs these inter-octamer interactions in solution or in cells.

We revised that sentence, as suggested (line 259).

8. Lines 307-308: It would be interesting to add purified N-terminal arm domain only to the in vitro reactions.

We will consider this experiment as part of our future studies of σ NS, although we wonder whether the purified N-terminal arm domain will fold natively when expressed alone. As we have no way to detect native folding of this short sequence, it will not be possible to interpret a negative result.

9. Line 321: Please restate “our discovery” given that this activity has already been described, at least for avian σ NS.

We revised that sentence as follows, “The RNA chaperone activity of σ NS raises an important question about the stage in viral replication at which this activity is required.” (Lines 313-314)

10. Line 335: Please add the strain of reovirus used for σ NS proteins (ie., T3D?).

The strain used was type 3 Dearing (T3D), which has been added to the Methods (line 328).

11. Line 340: The methods seem to indicate that all protein expressions were done with 10 mg/mL Se-Met. Please rephrase.

We clarified the protein expression protocols, as requested.

12. Lines 345-346: Please add information about how the A280 was measured for the elution profiles in Fig. 1.

This information is provided in the legend for Figure S1.

13. Please add information about protein purification yield, SDS-PAGE analysis (if performed) and protein storage conditions.

We added this information to the text (line 334, 341).

14. There are no methods for the molecular dynamics (Fig. 2c) or the molecular docking experiments (Fig. 6d-e). Please add.

We added sections in the Methods for the molecular dynamics and molecular docking experiments (lines 374-384).

Other changes:

New references:

We added three new references (refs 1-3) in support of the first sentence in the introduction (lines 42-43).

Added reference (39) in line 419 for how we predicted stem-loop structures.

Added reference (40) in line 424 for how we measured band intensity.

Reviewers' Comments:

Reviewer #1:

Remarks to the Author:

The authors have done an excellent job of modifying the MS to make it more readable. I have no further comments or suggestions. It is quite interesting work.

Reviewer #2:

Remarks to the Author:

The authors fully addressed all the minor concerns I raised earlier, and I look forward to seeing their work published.

Reviewer #3:

Remarks to the Author:

The authors have addressed the majority of previous reviewer concerns. The paper will be a very nice addition to the field.